**Data Availability Statement:** All relevant data are within the paper and its Supporting information files and in the excel file, de-identified.

# Mental health of individuals infected with SARS-CoV-2 during mandated isolation and compliance with recommendations—A population-based cohort study

**Anja Domenghino**[1,2], **Hélène E. Aschmann**[1,3], **Tala Ballouz**[1], **Dominik Menges**[1], **Dominique Strebel**[1], **Sandra Derfler**[1], **Jan S. Fehr**[1], **Milo A. Puhan**[1]*

1 Epidemiology, Biostatistics and Prevention Institute (EBPI), University of Zurich (UZH), Zurich, Switzerland, 2 Department of Visceral and Transplantation Surgery, University Hospital Zurich, University of Zurich (UZH), Zurich, Switzerland, 3 Department of Epidemiology and Biostatistics, University of California San Francisco (UCSF), San Francisco, California, United States of America

* miloalan.puhan@uzh.ch

## Abstract

### Background

Isolation is an indispensable measure to contain the SARS-CoV-2 virus, but it may have a negative impact on mental health and overall wellbeing. Evidence on the isolation experience, facilitating and complicating factors is needed to mitigate negative effects.

### Methods and findings

This observational, population-based cohort study enrolled 1547 adults from the general population with SARS-CoV-2 infection reported to authorities between 27 February 2020 and 19 January 2021 in Zurich, Switzerland. We assessed the proportion of individuals reporting symptoms of depression and anxiety before, during and after isolation (by DASS-21), and queried worries, positive experiences, and difficulties. We analyzed the association of these outcomes with socio-demographics using ordinal regression. Additionally, we report free-text statements by participants to capture most important aspects of isolation. The proportion of participants affected by depression or anxiety increased during isolation from 10·0% to 17·1% and 9·1% to 17·6%, respectively. Ordinal regression showed that taking care of children increased the difficulty of isolation (OR 2·10, CI 1·43–3·08) and risk of non-compliance (OR 1·63, CI 1·05–2·53), especially in younger participants. A facilitating factor that individuals commonly expressed was receiving more support during isolation.

### Conclusion

Isolation due to SARS-CoV-2 presents a mental burden, especially for younger individuals and those taking care of children. Public health authorities need to train personnel and draw from community-based resources to provide targeted support, information, and guidance to individuals during isolation. Such efforts could alleviate the negative impact isolation has on

**Funding:** The Zurich SARS-CoV-2 Cohort study is funded through the Health Directorate of the Canton of Zurich and the Pandemic Fund of the University of Zurich. The study is also part of the Corona Immunitas research program, coordinated by the Swiss School of Public Health (SSPH+) and funded through SSPH+ fundraising, including funding by the Swiss Federal Office of Public Health, the Cantons of Switzerland (Basel, Vaud, and Zurich), private funders (ethical guidelines for funding stated by SSPH+ were respected) and institutional funds of the participating universities. HEA was also supported by an SNSF Early Postdoc. Mobility Fellowship (nccr – on the move). TB received funding from the European Union's Horizon 2020 research and innovation program under the Marie Skłodowska-Curie grant agreement No 801076, through the SSPH+ Global PhD Fellowship Program in Public Health Sciences (GlobalP3HS) of the Swiss School of Public Health. Study funders had no role in the study design, data collection, analysis, interpretation, or writing of this report.

**Competing interests:** The authors have declared that no competing interests exist.

the mental and physical health of individuals and ensure compliance of the population with recommendations.

## Introduction

Isolation is an indispensable measure to contain the SARS-CoV-2 virus and affects millions of people worldwide [1, 2]. However, isolation, even if short-term, may also have a negative impact. Social isolation and loneliness affect mental health and can even be a risk factor for mortality [3]. The effect of mandated short-term isolation of people with infection or mandated quarantine for close contacts has become of interest in 2020 as it now suddenly affects millions of people. There is only limited evidence available from earlier epidemics and pandemics [4–6]. Adverse effects of short-term isolation included symptoms of post-traumatic stress disorder (PTSD), depression, anxiety, exhaustion, insomnia and even reports of suicide in individuals who were under quarantine or isolation due to Severe Acute Respiratory Syndrome (SARS), Ebola, H1N1 or Middle East Respiratory Syndrome (MERS) [7]. The persistence of mental health symptoms beyond isolation could depend on the isolation experience itself as well as the presence of pre-existing psychiatric illness [8, 9].

Since the start of the pandemic, most countries have implemented restrictions like social distancing, lockdowns, and mandated isolation for individuals testing positive for SARS-CoV-2. Current literature has assessed the effects of these measures [10–12]. There is evidence on the burden of isolation, but previous studies were non-population-based, conducted through convenience samples and often used cross-sectional online surveys. These studies show that isolation leads to more depression, stress and anxiety compared to stay-at-home orders or quarantine (e.g., after recent travel) alone [13–15]. However, there is little to no evidence from population-based studies on the effect of isolation on the mental health of individuals diagnosed with SASRS-COV-2. Furthermore, there is a need for evidence on factors that facilitate or complicate isolation. Therefore, the aim of this study was to assess the mental health of individuals diagnosed with SARS-CoV-2 throughout their mandatory isolation period in Switzerland, to analyze their worries and any positive experiences, and to report difficulties they faced in complying with the official isolation measures.

## Methods

### Study design and population

This analysis is based on the Zurich SARS-CoV-2 Cohort study, an ongoing observational, population-based cohort study of individuals in the Canton of Zurich with polymerase chain reaction (PCR)—confirmed SARS-CoV-2 infection, prospectively registered on the International Standard Randomized Controlled Trial Number Registry (ISRCTN14990068) [16]. Participants were identified through the Department of Health of the Canton of Zurich, to which all diagnosed SARS-CoV-2 cases of the canton are reported. To be eligible, individuals had to be at least 18 years old, residing in the Canton of Zurich, able to follow study procedures and have sufficient knowledge of the German language.

We enrolled two populations of SARS-CoV-2 infected individuals. The first, henceforth referred to as "retrospectively recruited", included all eligible individuals who were diagnosed with a SARS-CoV-2 infection prior to the start of the study (i.e., between 27 February 2020 and 05 August 2020). These individuals were enrolled between 06 October 2020 and 26 January 2021, at a median of 7.2 months after their diagnosis. The second, referred to as

"prospectively recruited", included an age-stratified random sample of all eligible individuals diagnosed between 06 August 2020 and 19 January 2021. These individuals were enrolled upon or shortly after diagnosis.

We obtained electronic written consent from all participants. The study protocol was approved by the ethics committee of the Canton of Zurich (BASEC 2020–01739).

## Procedures and data collection

Informed consent, recruitment and data collection was handled through the Research Electronic Data Capture (REDCap) survey system hosted at the University of Zurich [17, 18]. All participants completed an electronic questionnaire upon enrolment. In the prospective recruited sample, the baseline questionnaire included questions on socio-demographics, medical history, details on the acute SARS-CoV-2 infection (e.g., reason for PCR-testing, symptoms, and their severity), and mental health status prior to their infection. This was followed by a second survey two weeks and one month after receiving the positive test result. These surveys included additional questions related to their isolation experience and current physical and mental health status. The retrospectively recruited participants received their first questionnaire at a median of 7.2 months (range 5.9 to 10.3 months) after their positive test result. Due to this time lag and likely recall bias, we did not include questions on mental health nor compliance during isolation for this retrospective population. Free-text comments were available in all sections of all questionnaires for participants to describe additional issues (For example: "If you want to talk more about your worries during isolation you can do that here") or give details should the prespecified answers not fit (For example: "Place you isolated in: Other (Please specify where)").

## Outcomes

The primary aim of this study was to describe the mental health burden of individuals in mandatory isolation due to COVID-19. Specifically, we evaluated symptoms of negative emotional states before, during and after the time of isolation with the German version of the 21-item Depression, Anxiety and Stress Scale (DASS-21) [19]. To evaluate worries, positive experiences, and difficulties during isolation, we used five-point Likert scale questions with scales ranging from "extremely worried" to "not worried at all", "strongly agree" to "strongly disagree" and "very difficult" to "very easy", respectively.

Secondary outcomes included circumstances of isolation (perception of being well-informed about the recommended measures and place of isolation), compliance with isolation measures (e.g., avoiding contact with others, hygiene measures, wearing a mask) as specified by the Federal Office of Public Health (FOPH) and specific difficulties in doing so [20].

## Statistical analysis

We used descriptive statistics to analyze participants' baseline characteristics and the above-mentioned outcomes. Continuous variables are presented as median with minimum and maximum; categorical or ordinal variables as frequencies (N) and percentages (%). Results are presented for the total study population except when data was only available for those prospectively recruited. Missing values are reported where applicable. Calculation of the DASS-21 scores followed official guidance [19] and is described in detail in previously published research from our group [21].

We assigned corresponding severity levels of depression, anxiety, and stress according to official guidance [19]. We described the proportion of participants belonging to any of the DASS-21 categories stratified by age, sex, and COVID-19 symptom severity. We excluded data

from the two-week and one month follow-up questionnaires if they had completed them more than 30 or 60 days after diagnosis, respectively, or if all questionnaires were completed at the same time.

Multivariable ordinal regression was used on three categorical outcomes—perceived difficulty of isolation, perceived information status on recommended measures and compliance with these measures—to assess potential influences of sociodemographic variables. Variable selection was based on previous literature and preliminary results from free-text comment analysis. For example, many participants reported problems with adherence due to living with and caring for children, which is why we included the variable into our model. All multivariable models included age, sex, living situation, and occupation. For the models assessing perceived difficulty of isolation and compliance with recommended measures, we additionally considered information status as a variable. We added education level to the models evaluating information status and compliance with recommended measures. The regressions for difficulty and compliance were once done with age and living with children as interaction and once as independent variables. Results from the regression analysis are reported as odds ratios (OR) with 95% confidence interval (CI) and corresponding p-value.

We conducted sensitivity analyses to ensure robustness of study findings. In the first sensitivity analysis, we evaluated the impact of the second pandemic wave and the measures associated with it on DASS-21 scores. For that purpose, we divided participants into two groups: the first group included all participants who received their positive test result before 15 October 2020 (i.e., beginning of the second wave in Switzerland) and the second group included all participants diagnosed after 15 October 2020 [22]. In a second sensitivity analysis, we only included participants who completed all questionnaires at all three timepoints. Third, we conducted separate multivariable ordinal regression analyses of perceived difficulty of isolation for each of the retrospectively and prospectively recruited populations.

All free-text comment fields in the different questionnaires related to isolation or the overall experience during the SARS-CoV-2 infection were screened by two of the co-authors (SD and DS) for statements that are relevant to this study. All comments concerning isolation or mental health status were selected for further analysis. Relevant comments were assigned to the preliminary categories of circumstances of isolation, positive and negative aspects, mental health burden, and various based on the quantitative themes of the analysis. After screening the first third of the comments, themes were discussed between SD, DS and AD and were re-evaluated. Specific words that came up repeatedly were also quantified through text search. All comments were then reread by AD who recategorized some of them if necessary. We report themes that were repeatedly mentioned or that were considered especially impactful or important from a public health perspective.

R version 4.0.4 (2021-02-15) was used for all analyses [23].

## Results

### Study population

We included 1547 participants in our study. Among individuals diagnosed with SARS-CoV-2 between 27 February and 05 August 2020 (retrospectively recruited), 1311 were eligible and 442 agreed to participate (participation rate 33·7%). Among individuals diagnosed from 06 August 2020 until 19 January 2021 (prospectively recruited), 3204 were eligible and 1105 individuals agreed to participate (participation rate 34·5%) (S1 File).

There were no considerable differences regarding the sociodemographic characteristics of the two sample populations (Table 1). Of the 369 individuals living with children, 12 (3·3%)

**Table 1. Characteristics of participants enrolled in the Zurich SARS-CoV-2 cohort study.**

| Sociodemographic Background | Prospectively recruited (diagnosed between 06 Aug 2020–19 Jan 2021) | Retrospectively recruited (diagnosed between 27 Feb 2020–05 Aug 2020) | Total |
|---|---|---|---|
| | (N = 1105) | (N = 442) | (N = 1547) |
| **Sex** | | | |
| Female | 565 (51.1%) | 218 (49.3%) | 783 (50.6%) |
| Male | 540 (48.9%) | 224 (50.7%) | 764 (49.4%) |
| **Age groups** | | | |
| 18–39 years | 344 (31.1%) | 169 (38.2%) | 513 (33.2%) |
| 40–64 years | 448 (40.5%) | 209 (47.3%) | 657 (42.5%) |
| 65+ years | 313 (28.3%) | 64 (14.5%) | 377 (24.3%) |
| **Age (years)** | | | |
| Mean [Min, Max] | 50.2 [18.0, 92.0] | 46.5 [17.0, 87.0] | 49.2 [17.0, 92.0] |
| Median [IQR] | 50.0 [35, 66] | 47.0 [33, 57] | 49.0 [34, 63] |
| **Self-reported COVID-19 symptom severity** | | | |
| Asymptomatic | 148 (13.4%) | 47 (10.6%) | 195 (12.6%) |
| Mild | 248 (22.4%) | 69 (15.6%) | 317 (20.5%) |
| Moderate | 473 (42.8%) | 154 (34.8%) | 627 (40.5%) |
| Severe | 184 (16.7%) | 120 (27.1%) | 304 (19.7%) |
| Very Severe | 39 (3.5%) | 46 (10.4%) | 85 (5.5%) |
| Missing | 13 (1.2%) | 6 (1.4%) | 19 (1.2%) |
| **Comorbidities** | | | |
| One or more comorbidity | 324 (29.3%) | 148 (33.5%) | 472 (30.5%) |
| Missing | 9 (0.8%) | 8 (1.8%) | 17 (1.1%) |
| **Education** | | | |
| None or mandatory school | 45 (4.1%) | 21 (4.8%) | 66 (4.3%) |
| Vocational training and specialized baccalaureate | 459 (41.5%) | 188 (42.5%) | 647 (41.8%) |
| Higher technical school or college | 289 (26.2%) | 106 (24.0%) | 395 (25.5%) |
| University | 295 (26.7%) | 117 (26.5%) | 412 (26.6%) |
| Missing | 17 (1.5%) | 10 (2.3%) | 27 (1.7%) |
| **Job** | | | |
| Employed | 610 (55.2%) | 286 (64.7%) | 896 (57.9%) |
| Self-Employed | 107 (9.7%) | 41 (9.3%) | 148 (9.6%) |
| Student | 56 (5.1%) | 16 (3.6%) | 72 (4.7%) |
| Retired | 275 (24.9%) | 64 (14.5%) | 339 (21.9%) |
| Unemployed | 28 (2.5%) | 19 (4.3%) | 47 (3.0%) |
| Family manager | 14 (1.3%) | 4 (0.9%) | 18 (1.2%) |
| Missing | 15 (1.4%) | 12 (2.7%) | 27 (1.7%) |
| **Monthly household income (Swiss Francs)** | | | |
| <6'000 | 356 (32.2%) | 134 (30.3%) | 490 (31.7%) |
| 6'000–12'0000 | 458 (41.4%) | 156 (35.3%) | 614 (39.7%) |
| >12'000 | 226 (20.5%) | 120 (27.1%) | 346 (22.4%) |
| Missing | 65 (5.9%) | 32 (7.2%) | 97 (6.3%) |
| **Circumstances of Isolation** | **Prospectively recruited (diagnosed between 06 Aug 2020–19 Jan 2021)** | **Retrospectively recruited (diagnosed between 27 Feb 2020–05 Aug 2020)** | **Total** |
| | (N = 1105) | (N = 442) | (N = 1552) |
| **Living Situation during Isolation** | | | |

*(Continued)*

**Table 1.** (Continued)

| | | | |
|---|---|---|---|
| Living alone | 161 (14.6%) | 68 (15.4%) | 229 (14.8%) |
| **Living with Children** | | | |
| Yes | 251 (22.7%) | 118 (26.7%) | 369 (23.9%) |
| **Number of Days in Isolation*** | | | |
| Median [Min, Max] | 10.0 [2.00, 25.0] | - | - |
| Missing | 276 (25.0%) | - | - |
| **Place of Isolation**** | | | |
| At home | 1050 (95.0%) | 385 (87.1%) | 1435 (92.8%) |
| At someone else's home | 20 (1.8%) | 12 (2.7%) | 32 (2.1%) |
| In the Hospital | 26 (2.4%) | 50 (11.3%) | 76 (4.9%) |
| At a social Institution | 1 (0.1%) | 2 (0.5%) | 3 (0.2%) |
| At a hotel | 2 (0.2%) | 0 (0%) | 2 (0.1%) |
| Other | 19 (1.7%) | 14 (3.2%) | 33 (2.1%) |
| Missing | 13 (1.2%) | 7 (1.6%) | 20 (1.3%) |
| **Received information sheet on recommended measures** | | | |
| Yes | 1033 (93.5%) | 344 (77.8%) | 1377 (89.0%) |
| No | 62 (5.6%) | 90 (20.4%) | 152 (9.8%) |
| Missing | 10 (0.9%) | 8 (1.8%) | 18 (1.2%) |
| **Perception of Information status** | | | |
| Very well informed | 412 (37.3%) | 148 (33.5%) | 560 (36.2%) |
| Well informed | 539 (48.8%) | 189 (42.8%) | 728 (47.1%) |
| Neither poorly nor well informed | 104 (9.4%) | 69 (15.6%) | 173 (11.2%) |
| Poorly informed | 31 (2.8%) | 18 (4.1%) | 49 (3.2%) |
| Very poorly informed | 7 (0.6%) | 10 (2.3%) | 17 (1.1%) |
| Missing | 12 (1.1%) | 8 (1.8%) | 20 (1.3%) |
| **Know how to get additional information** | | | |
| Yes | 978 (88.5%) | 372 (84.2%) | 1350 (87.3%) |
| No | 114 (10.3%) | 64 (14.5%) | 178 (11.5%) |
| Missing | 13 (1.2%) | 6 (1.4%) | 19 (1.2%) |

*Only assessed in the prospectively recruited cohort,

** More than one answer possible, some participants isolated at two different locations

Table 1 displays the sociodemographic background of all participants, stratified by enrollment and time of infection. It presents circumstances of isolation like living situation, where the isolation took place, and how well participants felt informed about the recommendations.

were over 65 years old, 232 (62·9%) between 40 and 64 years and 125 (33·9%) between 18 and 39 years old (S1 File). We did not collect information on the age of the children.

## Circumstances of isolation

Most participants reported spending mandated isolation at home (Table 1). Fifty-five participants isolated at more than one place, in most cases (n = 41, 74·5%) first at the hospital and then at home. Thirty-three participants isolated at a holiday apartment or military hospital ward.

Overall, most (83·3%) participants reported feeling well informed about the recommended measures. However, participants from the retrospectively recruited sample felt less informed than those in the prospective sample (86·1% vs 76·3%) (Table 1). In the multivariable ordinal

regression, we found evidence that participants with children felt less well informed (OR 1.42, 95% CI 1.12–1.79) (S1 File). Only 18 participants described themselves as family managers (meaning primary care givers of children and household), and we found weak evidence that family managers also were less well informed (OR 2·11, 95% CI 0·89–4·98).

## Mental health during isolation

During mandated isolation, a higher proportion of participants experienced symptoms of depression (17·1%) and anxiety (17·6%) compared to before isolation (10·0% and 9·1%, respectively) or one month after the positive test result (14·7% and 14·8%). However, the proportion of participants suffering from severe symptoms of negative emotional states did not show a similar decrease after isolation ended for depression (from 1·6% to 2·8% to 2·6%), anxiety (from 1·8% to 3·6% to 4·0%), or stress (from 1·1% to 1·5% to 2·0%) at baseline, during isolation and one month after positive test result respectively (Fig 1). The percentage of those affected with stress increased during isolation as well but remained elevated after isolation. Subgroup analysis of male and female participants showed a smaller increase in males of 4·4% for symptoms of depression and 5·5% for symptoms of anxiety compared to an increase twice as large in females of 9·6% for depression symptoms and 11·3% for anxiety symptoms. Among participants suffering from mild symptoms of depression or anxiety during isolation, most (74·6% and 76·2%) reported not experiencing these symptoms at baseline (S1 File). Furthermore, we found that the proportion of participants suffering from symptoms of depression, anxiety, or stress during the isolation increased with the severity of COVID-19 symptoms in the acute phase and was highest among those with severe to very severe symptoms (Fig 1).

We found similar results in a sensitivity analysis comparing participants infected before or after 15 October 2020, and in another sensitivity analysis, restricted to participants who completed questionnaires at all three timepoints (931/1105 (84·3%) participants) (S1 File).

## Worries and positive aspects

Participants were most concerned about their loved ones (Fig 2a), about them becoming severely ill or getting infected with SARS-CoV-2, whereas concerns about job security and financial difficulties affected only few. In a subgroup analysis according to age strata (S1 File), participants aged between 18–39 years were more often very or extremely worried than those over 65 years of age in all domains of surveyed worries. A higher proportion of participants with pre-existing symptoms of depression, anxiety, or stress (217/352, 61·5%) were worried about the negative impact of isolation on their mental health compared to those without pre-existing symptoms (334/1167, 28·6%) and the total population (557/1547, 36%). Participants who live with children were more worried about stress (173/369, 46·9% vs. 432/1178, 36·6%), financial difficulties (72/369, 19·5% vs. 174/1178, 14·8%), and access to medical care (132/369, 35·8% vs. 377/1178, 32%) compared to those without children. Those living alone worried more about being lonely (114/229, 49·8% vs. 508/1318, 38·5%) and losing motivation (127/229, 55·5% vs. 654/1318, 49·6%) but less about stress (80/229, 21·8% vs. 525/1318, 39·8%) and infection or illness of others (161/229, 70·3% vs. 1056/1318, 80·1%). Out of the eight suggested positive effects of isolation, participants most often chose time spent at home and the opportunity to relax (Fig 2b). Participants over 65 years were less likely to find enjoyable parts in isolation compared to those younger (S1 File). Although those living with children found less time to relax, 57·5% reported appreciating being close to family.

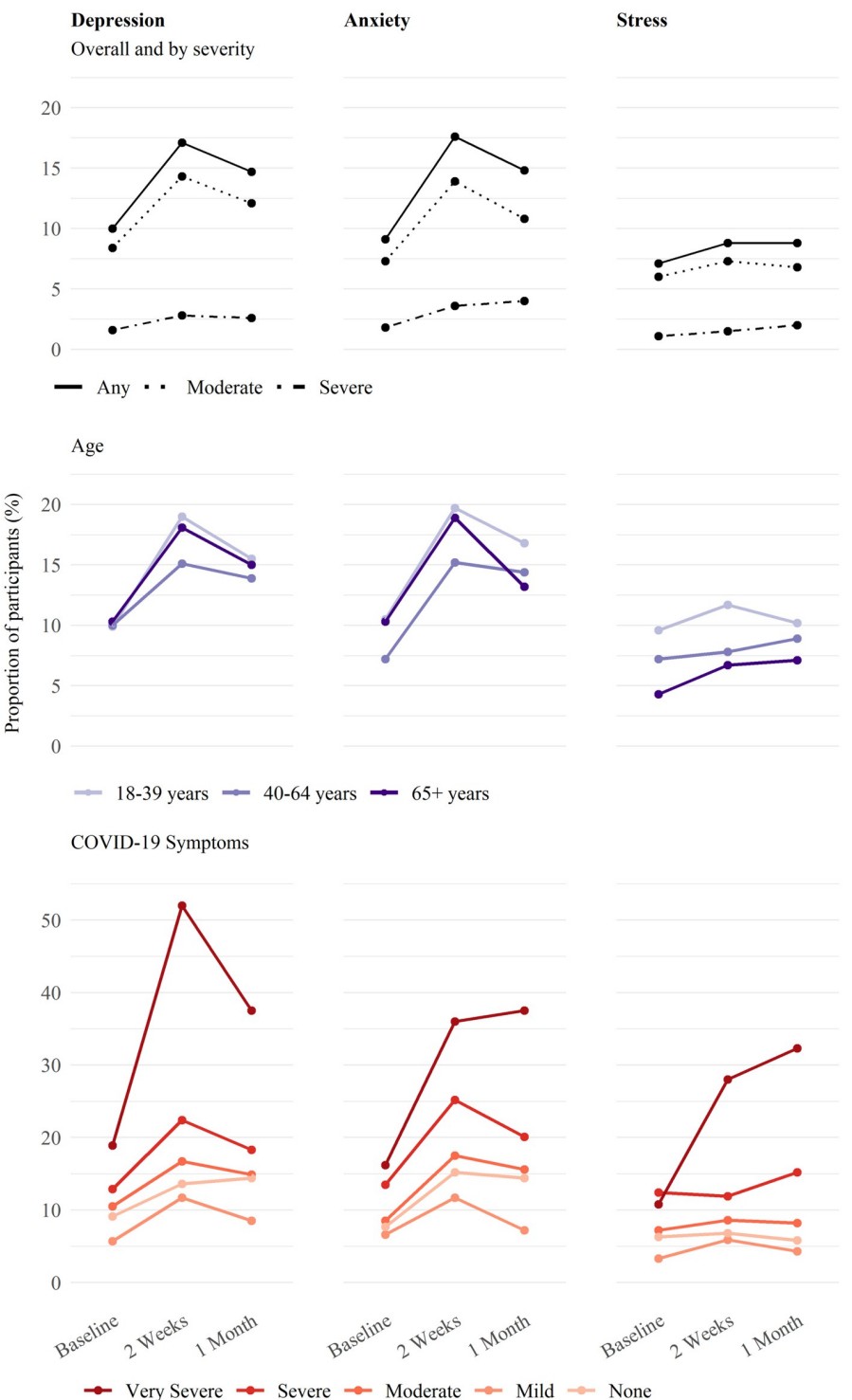

**Fig 1. Mental burden of isolation—Negative emotional states in the first month after positive SARS-CoV-2 test result.** Proportion of the population reporting symptoms of depression, anxiety, or stress through self-reported questionnaire at 3 different timepoints; before being diagnosed with SARS-CoV-2 infection, during isolation and 30 days after diagnosis, when isolation has ended. Results are reported overall and by symptom severity (i.e., moderate or severe), as well as stratified by age groups and subgroups of self-reported COVID-19 symptoms at infection. Out of 1105 participants, 1095 (99%) completed the baseline questionnaire, 964 (87%) completed the week two and 1050 (95%) the one-month questionnaire. We excluded 26 (2·3%) individuals due to large time differences between the positive test result and completion of the baseline questionnaire. The week two questionnaire was not available for 141 participants due to late recruitment.

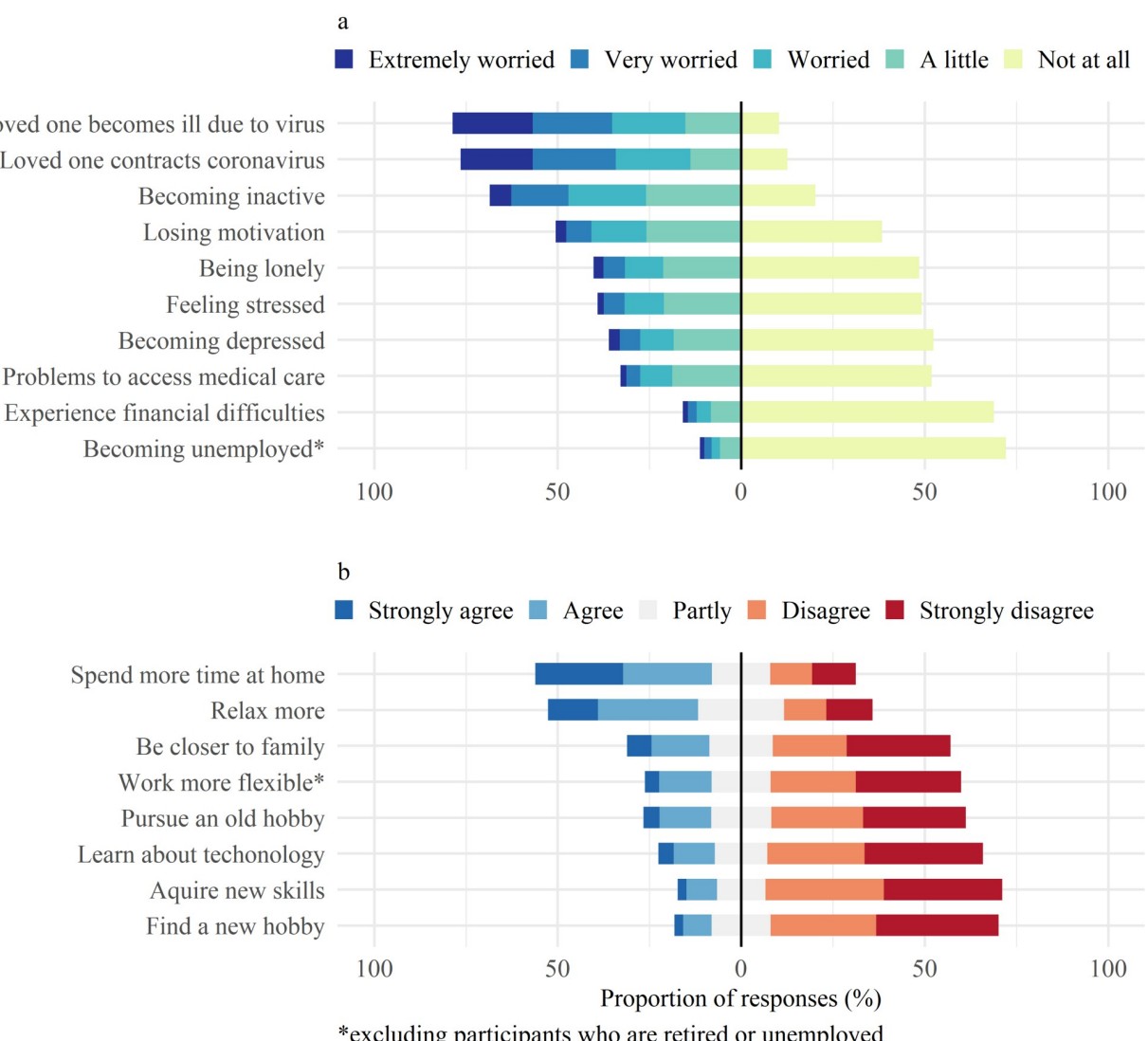

**Fig 2. Mental burden of isolation—Worries and positive aspects of isolation.** a) visualizes 10 worries of isolation prespecified in the questionnaire and the percentage of the population affected with different levels of concern asked in Likert scale from extremely worried to not worried at all. b) visualizes 8 possible positive effects of isolation prespecified in the questionnaire and the percentage of the population who perceived those as such, asked in Likert scale from strongly agree to strongly disagree.

### Difficulties of isolation and compliance with recommended measures

Participants were asked how difficult it was for them to comply with the recommendations of the FOPH overall and separately for each recommended measures during isolation. While 836 (54%) saw little or very little difficulty in complying with recommendations overall, avoiding contact with pets or staying alone in one's room was considerably harder (S1 File). Participants who live with children had twice the odds of experiencing difficulties with adherence to isolation recommendations when compared to those who do not live with children (OR 2·10, 95% CI 1·43–3·08). This applies especially to younger parents, as participants aged between 18–39 years had more difficulty in interaction analysis compared to those aged 40–64 who lived with children (Table 2). Participants who felt that they were poorly informed about the measures were also more likely to experience difficulties with compliance (OR 3·05, 95% CI 1·91–4·87).

**Table 2. Influence of sociodemographic background on perceived difficulty of Isolation in multivariable ordinal regression.**

| Total Cohort | | | |
|---|---|---|---|
| | **Odds Ratios** | **95% CI** | **p-value** |
| Male Sex | 0.74 | 0.61–0.89 | 0.002 |
| Living Alone | 0.67 | 0.51–0.89 | 0.007 |
| Living with Children | 2.10 | 1.43–3.08 | <0.001 |
| Living with Pets | 1.13 | 0.90–1.41 | 0.289 |
| **Age group** | | | |
| **Reference: 18–39 years old** | | | |
| Age: 40–64 years old | 0.84 | 0.65–1.10 | 0.210 |
| Age: 65+ years old | 0.74 | 0.45–1.24 | 0.259 |
| **Information Status** | | | |
| **Reference: Well informed** | | | |
| Neither poorly nor well informed | 1.70 | 1.27–2.28 | <0.001 |
| Poorly informed | 3.05 | 1.91–4.87 | <0.001 |
| **Occupation** | | | |
| **Reference: employed** | | | |
| Self-employed | 1.00 | 0.71–1.42 | 0.985 |
| In Education | 0.95 | 0.61–1.48 | 0.811 |
| Retired | 0.82 | 0.50–1.35 | 0.436 |
| Without Work | 0.96 | 0.57–1.61 | 0.880 |
| Family Manager | 1.26 | 0.53–3.00 | 0.604 |
| **Interaction with age (ref: 18–39) when living with children** | | | |
| 40–64 years old | 0.61 | 0.38–0.99 | 0.046 |
| 65+ years old | 0.38 | 0.12–1.15 | 0.089 |
| Observations | 1491/1547 | | |
| R2 Nagelkerke | 0.127 | | |

Table 2 describes association of sociodemographic background with the perceived difficulty level of adherence to isolation recommendations overall in all 1493 participants who stated their overall difficulty level.

We found similar results when conducting the analysis separately for each of the retro- and prospectively recruited population groups (S1 File).

We also analyzed difficulty to comply to isolation measures in the prospectively recruited cohort. A total of 901 (81·5%) participants reported adherence to isolation always or almost always. There was again an association with the living situation where individuals living with children or pets experienced more difficulties to comply (OR 1·63, 95% CI 1·05–2·53 and OR 1·43, 95% CI 0·94–2·15, respectively). Furthermore, there was weak evidence for those living alone to experience less difficulties to comply (OR 0·60, 95% CI 0·28–1·17) and for those who were self-employed to experience more difficulties to comply (Table 3). Different reasons for seeking testing were not associated with the level of compliance after the confirmed positive result. However, there were differences in adherence to quarantine before the confirmation of SARS-CoV-2 infection (S1 File). Only 75% of participants complied with quarantine recommendations always or almost always when the reason for getting tested was being symptomatic while it was 82·3% in those who were getting tested for any other reason.

We considered participants as non-compliant if they reported complying with the recommended measures "frequently" or less. A total of 156/1105 participants were considered non-compliant with the measures before their positive test result and 38/1105 were deemed so

**Table 3. Influence of sociodemographic background on complying to Isolation recommendations in multivariable ordinal regression.**

| Prospective Cohort | | | |
|---|---|---|---|
| | **Odds Ratios** | **95% CI** | **p- value** |
| Sex: male | 0.90 | 0.61–1.30 | 0.557 |
| Living Alone | 0.60 | 0.28–1.17 | 0.159 |
| Living with Children | 1.63 | 1.05–2.53 | 0.029 |
| Living with Pets | 1.43 | 0.94–2.15 | 0.090 |
| **Age group** | | | |
| **Reference: 18–39 years old** | | | |
| Age: 40–64 years old | 0.95 | 0.59–1.55 | 0.831 |
| Age: 65+ years old | 1.06 | 0.41–2.61 | 0.897 |
| **Information Status** | | | |
| **Reference: Well informed** | | | |
| Neither poorly nor well informed | 0.90 | 0.42–1.75 | 0.775 |
| Poorly informed | 1.25 | 0.45–2.94 | 0.638 |
| **Education** | | | |
| **Reference: none or mandatory** | | | |
| Vocational training and specialized baccalaureate | 1.24 | 0.45–4.37 | 0.706 |
| Higher technical school or college | 1.24 | 0.44–4.46 | 0.711 |
| University | 1.22 | 0.44–4.38 | 0.726 |
| **Reference: employed** | | | |
| Self-employed | 1.73 | 0.93–3.10 | 0.074 |
| Education | 1.54 | 0.67–3.29 | 0.287 |
| Retired | 0.96 | 0.38–2.45 | 0.924 |
| Without Work | 0.35 | 0.02–1.78 | 0.314 |
| Family Manager | 0.74 | 0.11–2.88 | 0.701 |
| Observations | 932/1105 | | |
| R2 Nagelkerke | 0.068 | | |

Table 3 describes association of sociodemographic background with the compliance to isolation recommendations overall in all 932 participants who stated their compliance.

afterwards. We observed higher proportions of non-compliance before the positive test result, among those aged 18 to 39 years (70/344, 20·4% compared to 49/448, 10·9% in those aged 40–64 years and 37/313, 11·2% in those over 65 years), living with children (16.0% compared to 13·6% in those who do not live with children), and females (86/565, 15·2% compared to 70/540, 13.0% in males). A higher proportion of participants went to mandatory school or had no degree in the group that were "non-compliant" after the positive test result when compared to the total population (10·5% vs 4·1%) and more were living with children (14/251, 5·6%) when compared to those living without children (24/854, 2.8%). There were no age or sex differences.

The prospectively enrolled cohort was additionally surveyed on pre-defined practical challenges they experienced while in isolation (Fig 3). Overall, participants found it most difficult to stay in good physical shape and to be alone, with a higher proportion of participants with difficulties in the youngest age group (S1 File). Maintaining harmony at home was more challenging for those raising children, where 74 (29·5%) found this difficult, compared to 169 (15·3%) in the overall population. There were no differences between males and females.

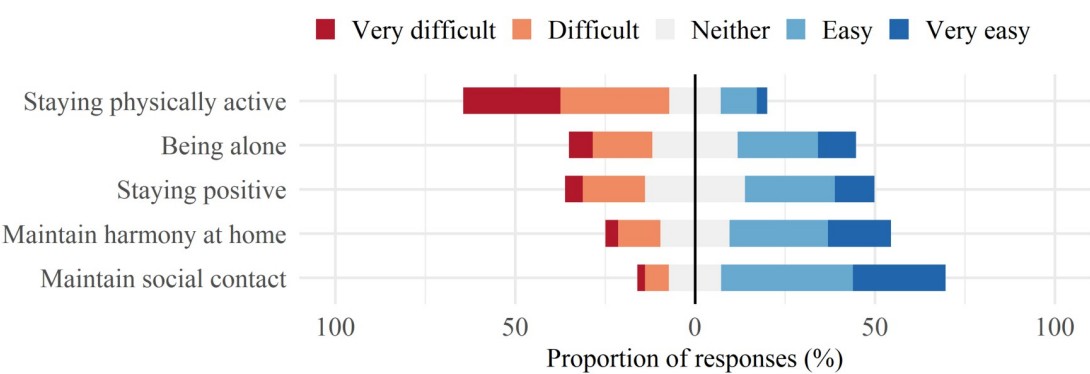

**Fig 3. Specific difficulties of isolation—Problematic aspects when undergoing the time in confinement.** Visualizes the percentage of the population affected by 5 difficult aspects of isolation prespecified in the questionnaire, on Likert scale from very difficult to very easy.

## Results—Free text comments—Separate box or supplementary materials

### Mental health during isolation

Feelings of loneliness, depression, and aggression because of isolation occurred frequently. Isolation was described as "prison", "solitary confinement" and even "torture" or "punishment" in over 23 comments, and some individuals felt like a "criminal" or "perpetrator" more than 13 times. There were participants who stated the time in isolation had "degraded their mental health" or "made them feel depressed and sad". Those already in a difficult emotional or psychological situation felt their situation degrade through isolation. Some stated that "it worsened preexisting depression" and some indeed feared that they "need to be institutionalized". Not knowing the consequences of the SARS-CoV-2 infection itself added to mental health issues, some individuals indicated they spent the time in isolation in "panic" and "mortal fear" and stated "I asked myself all the time; will I die today?","will I ever see my family again?", "I was scared I would die, and nobody would notice", and "being alone intensified my fears".

Participants described somatic pain due to lack of physical activity or absence of social contact: "the isolation was psychologically and physically draining, tiring and discouraging", "after isolation I needed a few days to restart my body, because of the missing exercise I experienced backpain". They reported loss of appetite and depression as a direct consequence of isolation. They stated that the more time they spent in isolation, the more physically ill they felt. More than ten participants stated that "isolation was more wearing than the illness itself", "for me the worst was the isolation, it was mentally and physically exhausting, tiring and discouraging", "the isolation was a thousand times worse than the infection itself". Or as one individual summarized it: "I felt sad, depressive, alone, negative and tainted, it was just really hard". Having the possibility to leave the house for a walk or some time in nature would have been a great relief for many.

### Worries and positive aspects

Participants were generally very worried about individuals they might have infected including loved ones. However, they also worried about facing stigma and consequences at the workplace. They perceived that coping with isolation was more manageable if their employer handled the situation well. individuals stated "being blamed for getting infected by my boss" or

"my supervisor told me I should still come to work although I felt quite sick" which generated additional stress during isolation. Internal communication to co-workers was either perceived as "name and shame under the disguise of transparency" or the "aim of keeping my identity a secret lead to irritating situations". A person stated: "I felt like a criminal, especially at work". Whenever employers were "understanding that I am not fit for work yet" or "called every day to ask how I was", employees described the isolation as less stressful and even positive. Participants working in the health care system worried about abandoning their co-workers in this time of need and some also mentioned not feeling well protected at their workplace. While worries about financial troubles and job security were not frequent, they were highly stressful for those affected by them: "I could become a welfare case very soon", "I am bankrupt and unemployed due to COVID-19".

Still, some participants mentioned positive aspects to the isolation, mainly spending time with family and relaxing more often: "Our family became closer", "I saw that we as a couple were resilient under duress","The isolation was a pleasant opportunity to slow down". The word "deceleration" appeared frequently. Although individuals welcomed the possibility of video phone calls, they mentioned that "Phone calls and WhatsApp were no substitute for real human contact, but they helped for a short time". Some participants felt their time in isolation made them "refocus" on important things in life and "appreciate what they have". "The isolation gave me time to reorder my priorities, it was a wake-up call". There were participants who organized their homes and computers, learned a new language, or found a new hobby, did yoga and home workouts, and found leisure time to read books and listen to podcasts. "It was an exciting experiment, I was comfortable in my room, did gymnastics every day with video tutorials, had time for me to read, to work, and just be." However, this was only possible for those in good shape and having a strong support system, "I was too weak and mentally unstable to use this time positively".

## Difficulties of isolation and compliance

The challenge of trying to adhere to isolation recommendations while simultaneously caring for a family with little children was mentioned in multiple statements:"Emotionally the situation with my family was the hardest, I am alone in one room, my wife must take care of the two children, and no one can leave the house", "It was really hard for the children, they feared for my life". Participants were especially conflicted between home office and childcare and reported difficulties in explaining to little children why separation was necessary: "As parents we do not want to infect our children but at the same time have to care for them, this was the greatest psychological stress for our family". Having little children undergo isolation or quarantine was also mentioned as impossible:"completely unsuitable was the isolation for my two children (three and five years old), to lock them in one room for ten days is absurd and not appropriate for little kids". Families demanded more adjusted measurements, help and alternatives for their individual needs. Relationships suffered as well during the time in isolation: "It was a burden for our relationship", "my partner wants me to abstain from any social contact", "I react verbally aggressive because I cannot leave my apartment".

One of the biggest issues for participants was that they felt not well informed about when it was safe to leave isolation and meet people afterwards, "I missed care and assistance during the time after isolation". Differing opinions about that in the social environment created psychological strain and led to conflicts. Participants felt especially dissatisfied and alone when they received contrary instructions from different authorities (e.g., contact tracing, private doctor, hospital): "It was mentally challenging to manage different information about the duration of isolation", "I had no information on how to behave after isolation", "friends did not want to

meet with me", "I was not allowed back to work because people were scared I could infect someone", "The FOPH says I can move freely, my private doctor is uncertain and the hospital does not let me enter because of the previous infection, I do not know what to think". Another regular concern was the communication with close contacts. When do you inform your contacts that you suspect you have COVID-19? This issue led to conflict and discussions in the private and work environment: "It would be helpful to always inform your surroundings of even mild symptoms so one has the possibility to protect themselves from infection".

There was also a great need for more follow-up and assistance through the government in general, the diagnosing physician or any medical personal during isolation: "I felt abandoned by the government", "Not one person asked about me as a patient, how I was", "I was appalled by the poor care I received as a sick person, I am an at-risk patient and felt very alone", "During the whole ten days not one medical professional asked about my health", "the available assistance did not help us find realizable solutions, we felt unescorted", "I would have wished for support from the government during this time", "I needed some exchange with a doctor, to answer my questions as I did not want to go to the hospital with small issues.", "No one felt responsible for treating me, my private doctor was not here, their replacement did not want to take me on and for the hospital I was not sick enough". People also wished for more direction on how to treat the symptoms of their illness during the time in isolation.

Participants mentioned they could benefit from private exchange with other affected people: "I had an acquaintance who also was infected during the same time and our active and close interchange helped me so much", "I had the need to exchange my experience with other positive tested individuals". But also, how this might be stressful, "I became the info hotline for all COVID-19 related questions, it drives me mad". Participants felt alone and unescorted during this time which made them feel frustrated and helpless.

## Impressions from free text comments (should full comments be in the supplementary material)

Throughout the free text comments, participants expressed feelings of loneliness and depression as a consequence of their isolation. Isolation was described as "prison", "solitary confinement" and even "torture" or "punishment" in over 23 comments. Participants described somatic pain due to the lack of physical activity or absence of social contact and described loss of appetite. More than ten participants stated in some form that "isolation was the worst part of their illness". Participants were most worried about individuals they might have infected, especially loved ones. They also worried about facing stigma and consequences at the workplace. While worries about financial troubles and job security were not frequent, they were highly stressful for those affected by them. Positive comments mainly mentioned spending time with family and relaxing more often. The word "deceleration" appeared frequently.

The challenge of trying to adhere to isolation recommendations while simultaneously caring for a family with little children was one of the most striking impressions from statements. Participants were especially conflicted when children had to undergo isolation or quarantine themselves:"*completely unsuitable was the isolation for my two children (three and five years old), to lock them in one room for ten days is absurd and not appropriate for little kids*". Families demanded more adjusted measures, help and alternatives for their individual needs. Overall, the call for more and better information, for example about when it was safe to leave isolation and meet people afterwards, was high. There was also a great need for more follow-up and assistance through the government in general, the diagnosing physician or any medical personnel during isolation: *"During the whole ten days not one medical professional asked about my health"*, *"the available assistance did not help us find realizable solutions, we felt unescorted"*.

These feelings of loneliness and perception of lack of support during this time which made them feel frustrated and helpless.

## Discussion

### Key findings

In this observational, population-based cohort study, we described the burden of isolation on SARS-CoV-2 infected individuals in Zurich, Switzerland and identified especially vulnerable individuals. We found that symptoms of depression and anxiety increased during mandated isolation, particularly in younger participants and in those with severe symptoms of COVID-19. We additionally observed that caring for children increased the difficulty of isolation and risk of non-compliance especially in younger participants. Importantly, participants consistently highlighted the need for clear guidance and support during and after the isolation period.

While some of the increase in symptoms of depression and anxiety could potentially be attributed to symptoms of COVID-19 (particularly fatigue) and the following decrease therefore to their recovery, the pattern is visible even in those with mild and moderate symptoms. Furthermore, while depression and anxiety receded after isolation, symptoms of stress remained. These findings align with previous studies on previous epidemics or pandemics describing depression, anxiety, and stress, in quarantine isolation [5]. These were findings from single observational studies, reporting on either one or the other and did not focus on people with a SARS-CoV-2 infection. The numerous free-text comments gave additional important insights into the experience of isolation. Especially concerning were remarks on how the isolation itself seemed to intensify symptoms of pre-existing mental health illnesses, of COVID-19 or even induce somatic pain and symptoms. Over 60% of individuals with pre-existing symptoms of depression were concerned about their mental health during isolation and the percentage of those with severe depression, anxiety or stress rose continuously during the month following infection with SARS-CoV-2. Participants already under emotional, psychological, or existential stress were affected most severely and had existential fears, although worries about job loss and money were the least frequent during isolation. Our observations and those of others suggest that contact tracing teams should be alert for suicidal ideations in individuals with pre-existing mental health problems [13, 24].

Taking care of children had a significant impact on the difficulty of isolation, the ease to comply with regulations and even the perceived information status of participants. Parents had more problems maintaining harmony at home, which aligns with reports of more domestic violence reported all over the world [25]. Especially those between 18–39 years of age had more difficulties when compared to those older than 40 years, which is likely due to the younger age of their children who need even more attention. Unfortunately, we did not elicit the age of the children. However, this assumption was also supported by participant comments where those over 40 years reported positive experiences with family and mentioned conversations and activities mostly reserved for older children. Whether with or without children, overall, younger participants seemed to be most affected by isolation. They experienced more difficulties during this time and were worried more than the older age groups. However, they were also more likely to experience positive aspects of isolation. Their needs during isolation differed considerably from older age groups and need to be taken seriously, especially as they were also almost twice as often "non-compliant" before their positive test result, when compared to the older population. Most participants strictly adhered to the rules of mandated isolation and reported little difficulty overall. While there might be some reporting bias present and participants in the cohort study were probably more compliant than the general

population, there were still many who did not follow regulations consistently. This gives us important insights into their demographic background and reasoning for non-compliance.

## Public health implications and further research

Participants reported requiring closer monitoring and additional support for their physical and mental wellbeing during isolation. Public health authorities should implement more standardized follow-up of individuals infected with SARS-CoV-2 and train personnel of contact tracing or government staff to better identify individuals in need of help. The support, however, must not solely be provided by government facilities or medically trained personnel. While medical and psychological assistance should be offered by experienced doctors and uncertainties related to regulations are usually best addressed by healthcare officials, there are still opportunities for creative solutions to support individuals by friends, family, or community workers and volunteers. Offering services for physical online activities in group sessions could alleviate issues on multiple levels. Programs like HOMEX have shown that it is possible also for those over 65 years of age to do at home training sessions through online videos [26]. Offering group online activities would enable infected individuals to stay active and to be in safe contact with other individuals in the same situation. This is something that participants suggested as helpful in free-text comments and in previous studies [4]. While follow-up should be standardized, solutions must be more personalized. Our data showed that caring for children and especially working exclusively as a family manager significantly and negatively impacts the perceived information level and the difficulty of complying with isolation. Supporting young families in providing information and helping them find targeted solutions for childcare or isolation measurements must be a priority for healthcare services to ensure compliance and reduce negative effects on mental health of parents and children. Recommendations for online educational platforms, targeted to specific populations, have been made but not yet applied [27]. Possibilities in online group and video support are vast and should be utilized more effectively.

The other main request from participants were clear guidelines, especially on leaving isolation. Most individuals worry most about infecting others, and the duration of confinement is one of the main stressors of isolation [4]. Therefore, keeping it as short as possible should be of upmost importance. Healthcare officials could offer contact points where infected individuals can discuss their questions about the safety of leaving isolation and use this as reference point for their personal and professional environment. Precise guidelines could also improve work-related issues. Results from our cohort highlighted that the attitude of the employer and the work environment influences the isolation experience of employed individuals heavily. Precise and applicable guidelines for employers that protect infected staff and allow for some recreation time would provide relief for employees, possible contact points to clarify particular issues could help settle conflicts. Furthermore, guidelines and especially further information could help with compliance, since it is a cause for concern that a quarter of participants who were tested because of symptoms of COVID-19 did not comply with stay-at-home orders more than frequently. Considering a possible selection and reporting bias, the true number of non-compliant individuals may be even higher. This suggests that individuals need more incentives to stay at home before receiving a positive test result and public health authorities should use this information to remind especially the younger age groups of the importance of complying with measures.

Further research should focus on the development and evaluation of targeted support for individuals most affected by the time in isolation. Especially innovative solutions for families, multigenerational households and for those living alone or with mental and existential issues

are needed. Assistance should not necessarily be carried out by government personnel. Creative solutions like support through friends, other affected or recovered individuals, community workers or volunteers are warranted. However, such offers must be coordinated through and briefed by official channels. The importance of rapid and effective communication through health care officials has been highlighted before [4]. However, it is crucial that conversation goes both ways. Information flows need to be improved not only to support individuals during this challenging time but also to ensure that personnel who are at the first point of contact with individuals are adequately trained to identify and support individuals in need. Programs evaluating how to improve pandemic preparedness should include better measures, support during isolation, and highlight the need to actively involve affected individuals in the development of support programs.

## Limitations

This study has several limitations. First, participants in the retrospectively recruited sample were diagnosed during the first wave in Switzerland during which testing was limited to older individuals or those who were at risk of severe disease. This population is thus older and burdened with more symptoms and may not be representative of the full range of the infected population. On the other hand, the prospectively recruited sample captures the range of the population more adequately. Second, participants who participated may have been more likely to be concerned with their health and more compliant with the measures. This and social desirability bias probably could have led to an underrepresentation of individuals not complying with recommendations of the FOPH. However, the overall focus of the Zurich SARS-CoV-2 Cohort study is to analyze the development of the disease and immune response over time, so individuals did not participate to recount their isolation experience. Third, we did not evaluate the age of children living in the same household and therefore had to use the age of participants as a substitution. However, the impact of having children on the isolation experience was present throughout free-text comments and stayed robust in all sensitivity analyses. Last, the methodological approach of analyzing Likert scale items through ordinal regression has its limits, and the factors included in our two models explained only a small proportion of the variance in compliance with isolation and perceived difficulty as seen by Nagelkerke's $R^2$. While we found consistent results such as increased difficulty for participants with children and differences between the age groups throughout all analyses and in all the visual presentations, further research is needed to investigate which other factors influence these outcomes.

## Conclusion

This observational, population-based cohort study showed that younger individuals and those who are living with children experience considerable mental health burden and encounter intense worries and difficulties during isolation due to COVID-19. Insufficient information on recommended measures during that period can be additionally challenging. Public health authorities need to offer targeted support, information, and guidance to individuals to relieve the negative impact isolation has on the mental and physical health of individuals and to ensure compliance of the population with public health measures.

## Supporting information

**S1 File.**
(DOCX)

**S1 Data.**
(XLSX)

## Acknowledgments

This study would not have been possible without the exceptional work and commitment of the study administration staff and the staff of the Corona Center of the University of Zurich. The authors would also like to thank the Department of Health of the Canton of Zurich for their support and collaboration in realizing this project. And above all else, the results of this study are owed to all the participants who dedicated their personal time to provide data the Zurich SARS-CoV-2 Cohort.

## Author Contributions

**Conceptualization:** Anja Domenghino, Hélène E. Aschmann, Tala Ballouz, Dominik Menges, Jan S. Fehr, Milo A. Puhan.

**Data curation:** Anja Domenghino, Tala Ballouz, Dominik Menges, Dominique Strebel, Sandra Derfler, Milo A. Puhan.

**Formal analysis:** Anja Domenghino, Hélène E. Aschmann, Tala Ballouz.

**Funding acquisition:** Tala Ballouz, Dominik Menges, Jan S. Fehr, Milo A. Puhan.

**Investigation:** Anja Domenghino, Hélène E. Aschmann, Tala Ballouz, Dominik Menges, Dominique Strebel, Sandra Derfler, Milo A. Puhan.

**Methodology:** Anja Domenghino, Hélène E. Aschmann, Tala Ballouz, Dominik Menges, Milo A. Puhan.

**Project administration:** Anja Domenghino, Tala Ballouz, Dominik Menges, Jan S. Fehr, Milo A. Puhan.

**Resources:** Tala Ballouz, Dominik Menges, Jan S. Fehr, Milo A. Puhan.

**Software:** Anja Domenghino, Hélène E. Aschmann, Tala Ballouz, Dominik Menges.

**Supervision:** Anja Domenghino, Tala Ballouz, Dominik Menges, Jan S. Fehr, Milo A. Puhan.

**Validation:** Anja Domenghino, Hélène E. Aschmann, Tala Ballouz, Milo A. Puhan.

**Visualization:** Anja Domenghino, Hélène E. Aschmann.

**Writing – original draft:** Anja Domenghino, Milo A. Puhan.

**Writing – review & editing:** Anja Domenghino, Hélène E. Aschmann, Tala Ballouz, Dominik Menges, Dominique Strebel, Sandra Derfler, Jan S. Fehr, Milo A. Puhan.

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
