## [Decision Letter · Decision Letter 0]

18 Jan 2022

PONE-D-21-39431Mental Health of Individuals Infected with SARS-CoV-2 during Mandated Isolation and Compliance with Recommendations - a Population-Based Cohort StudyPLOS ONE

Dear Dr. Milo A Puhan,

Thank you for submitting your manuscript to PLOS ONE. After careful consideration, we feel that it has merit but does not fully meet PLOS ONE’s publication criteria as it currently stands. Therefore, we invite you to submit a revised version of the manuscript that addresses the points raised during the review process.

In this review, two reviewers provided excellent comments. Please address all issues to polish your work.

We look forward to receiving your revised manuscript.

Kind regards,

Wen-Wei Sung, M.D., Ph.D.

Academic Editor

PLOS ONE

Journal Requirements:

Reviewers' comments:

Reviewer's Responses to Questions

**Comments to the Author**

1. Is the manuscript technically sound, and do the data support the conclusions?

Reviewer #1: Yes

Reviewer #2: Yes

2. Has the statistical analysis been performed appropriately and rigorously? 

Reviewer #1: Yes

Reviewer #2: Yes

3. Have the authors made all data underlying the findings in their manuscript fully available?

Reviewer #1: Yes

Reviewer #2: Yes

4. Is the manuscript presented in an intelligible fashion and written in standard English?

Reviewer #1: Yes

Reviewer #2: Yes

5. Review Comments to the Author

Reviewer #1: Important note: This review pertains only to ‘statistical aspects’ of the study and so ‘clinical aspects’ [like medical importance, relevance of the study, ‘clinical significance and implication(s)’ of the whole study, etc.] are to be evaluated [should be assessed] separately/independently. Further please note that any ‘statistical review’ is generally done under the assumption that (such) study specific methodological [as well as execution] issues are perfectly taken care of by the investigator(s). This review is not an exception to that and so does not cover clinical aspects {however, seldom comments are made only if those issues are intimately / scientifically related & intermingle with ‘statistical aspects’ of the study}. Agreed that ‘statistical methods’ are used as just tools here, however, they are vital part of methodology [and so should be given due importance]. To improve the article/presentation, clues/hints may be taken from this review but should not limit the process by adhering to those points alone.

COMMENTS: Although the study is very important, of current interest and the study is well conducted, I have few concerns. According to account given in lines 116-122 [We enrolled two populations of SARS-CoV-2 infected individuals. The first, henceforth referred to as “retrospectively recruited”, included all eligible individuals who were diagnosed with a SARS-CoV-2 infection prior to the start of the study (i.e., between 27 February 2020 and 05 August 2020). These individuals were enrolled between 06 October 2020 and 26 January 2021, at a median of 7.2 months after their diagnosis. The second, referred to as “prospectively recruited”, included an age-stratified random sample of all eligible individuals diagnosed between 06 August 2020 and 19 January 2021. These individuals were enrolled upon or shortly after diagnosis.], however, for analyses (example: application of ‘multivariable ordinal regression’ – table-2) you seem to have combined these samples. Even if (refer to lines 203-4) observing that “There were no considerable differences regarding the sociodemographic characteristics of the two sample populations (Table 1)”, is that {combining samples} correct? [Agreed that there were no considerable differences regarding the sociodemographic characteristics but the fact is these two samples collected by different methods]

In ‘abstract- Methods and Findings’ section, lines 33-35 only prospective cohort study [“This population-based prospective cohort study enrolled 1547 adults from the general population with SARS-CoV-2 infection] is mentioned. This mixing confusion is seen at other places also {example: Section on ‘Study design and Population’ (line 108 onwards) “This analysis is based on the Zurich SARS-CoV-2 Cohort study, an ongoing observational, population-based cohort study of individuals in the Canton of Zurich with polymerase chain reaction (PCR) -confirmed SARS-CoV-2 infection”. Please address this issue to clarify more. [No problem if it is so, but should be clarified, remember that this is a scientific/academic document and so all details should be clearly/correctly communicated]

In the context of information given in lines 144-46 [“To evaluate worries, positive experiences, and difficulties during isolation, we used five-point Likert scale questions with scales ranging from “extremely worried” to “not worried at all”, “strongly agree” to “strongly disagree” and “very difficult” to “very easy”, respectively], please note the following { pasted from one standard textbook on ‘Research Methodology’}:

Whenever response options ranged from 1=strongly disagree to 4=strongly agree (or ranging from 1 (strongly disagree) to 6 (strongly agree) or from 1=very bad to 3=neither good nor bad to 5=very good), while using a ‘Likert’ scale responses, recoding [like strongly disagree=-2, disagree=-1, neutral=0, agree=1, strongly agree=2] may yield correct and meaningful ‘arithmetic mean’ which is useful not only for comparison but has absolute meaning. Application of any statistical test(s) assume that meaning of entity used (mean, SD, etc) has a particular meaning. Though ‘α’ [alpha] or most other measures of reliability/correlation will remain same, however, use of non-parametric methods should/may be preferred while dealing with data yielded by any questionnaire/score.

Also consider that reported (observed) value of Nagelkerke’s R2 (=0.127) in table-2 is small and table-3 ( R2=0.068 ) is very very small. That the amount of variation explained is all most negligible. From (lines 157-8: Calculation and imputation of DASS-21 scores is described in previously published research from our study and followed official guidance (21).) it seems that the present sample is a sub-sample of some other study. If so, why this fact is not made clear in the beginning? Again, remember that this is a scientific/academic document and so all details should be clearly/correctly communicated.

Most of the statistical values/results are not interpreted adequately [example: Odds Ratios in table-3, male Sex (=0.90); Living Alone (=0.60)]. Study has potential, however, I suggest to consider above points.

Reviewer #2: The authors investigated impacts on the mental health due to the isolation experience in the general population with SARS-CoV-2 infection in Zurich, Switzerland. They found that the proportion of participants affected by depression or anxiety increased during isolation from 10·0% to 17·1% and 9·1% to 17·6%, respectively. They also found that taking care of children increased the difficulty of isolation (OR 2·10, CI 1·43 – 3·08) and risk of non-compliance (OR 1·63, CI 1·05 – 2·53), especially in younger participants. The topic seems to be interesting for researchers and practitioners in public health. In general, this is a well-designed, -analyzed and -written paper. I have some comments to improve this paper.

1. The description of the results about free text comments, i.e. Results - Free text comments - separate Box – (Line 344 to 435), is redundant for the readers. Results of free text comments should be summarized in one paragraph in the main manuscript. The original description of the results about free text comments should be presented as a supplementary material, if needed.

2. Clarification of the Figure 1 are needed. The lines of the all graphs are indistinguishable for the readers, e.g. the lines for “Any” and “Severe” in overall with severity, and the lines for “Very severe”, “Severe”, “Moderate” and “Mild” in COVID-19 Symptoms.

3. There are some typos in the manuscript:

Line 208, 55 participants …

Line 214, that that participants …

Line 252, when isolation has ended Results are …

Line 254, COVID-19 symptoms at infection Out of 1105 …

6. PLOS authors have the option to publish the peer review history of their article (what does this mean?). If published, this will include your full peer review and any attached files.

Reviewer #1: No

Reviewer #2: No

---

## [Author Response · Author response to Decision Letter 0]

3 Feb 2022

Point-by-point responses to reviewer comments (PONE-D-21-39431)

Reviewer #1: Important note: This review pertains only to ‘statistical aspects’ of the study and so ‘clinical aspects’ [like medical importance, relevance of the study, ‘clinical significance and implication(s)’ of the whole study, etc.] are to be evaluated [should be assessed] separately/independently. Further please note that any ‘statistical review’ is generally done under the assumption that (such) study specific methodological [as well as execution] issues are perfectly taken care of by the investigator(s). This review is not an exception to that and so does not cover clinical aspects {however, seldom comments are made only if those issues are intimately / scientifically related & intermingle with ‘statistical aspects’ of the study}. Agreed that ‘statistical methods’ are used as just tools here, however, they are vital part of methodology [and so should be given due importance]. To improve the article/presentation, clues/hints may be taken from this review but should not limit the process by adhering to those points alone.

We would like to thank Reviewer 1 for their insightful comments. We agree that statistical methods are a vital part of our methodology and should be give due importance. We made changes to the manuscript as listed below the individual comments. Line numbers refer to the manuscript without track changes.

COMMENTS: Although the study is very important, of current interest and the study is well conducted, I have few concerns. According to account given in lines 116-122 [We enrolled two populations of SARS-CoV-2 infected individuals. The first, henceforth referred to as “retrospectively recruited”, included all eligible individuals who were diagnosed with a SARS-CoV-2 infection prior to the start of the study (i.e., between 27 February 2020 and 05 August 2020). These individuals were enrolled between 06 October 2020 and 26 January 2021, at a median of 7.2 months after their diagnosis. The second, referred to as “prospectively recruited”, included an age-stratified random sample of all eligible individuals diagnosed between 06 August 2020 and 19 January 2021. These individuals were enrolled upon or shortly after diagnosis.], however, for analyses (example: application of ‘multivariable ordinal regression’ – table-2) you seem to have combined these samples. Even if (refer to lines 203-4) observing that “There were no considerable differences regarding the sociodemographic characteristics of the two sample populations (Table 1)”, is that {combining samples} correct? [Agreed that there were no considerable differences regarding the sociodemographic characteristics, but the fact is these two samples collected by different methods]

Thank you for highlighting this important point. While the two populations were selected similarly (i.e., as random samples of all cases identified by contact tracing in the Canton of Zurich), we agree that there are some differences, mainly relating to the time between infection and recruitment and the age-stratified random sampling of only the prospectively recruited population. We agree that in certain situations, the combination of population groups would not be appropriate regardless of the similarity of their sociodemographic characteristics. However, for our regression analyses on the overall difficulty on isolation, we judged that combining the two populations would be appropriate because we assume that the difficulty of the isolation experience is something that very memorable to participants and can be assessed quite similarly after some time. Nonetheless, we have additionally conducted a separate regression analysis for each of the populations and have found similar results. We have added this analysis to the methods section (lines 183-184) and provided the findings in the results section (lines 298-299) and supplementary material (Tables S5a and S5b). 

“Third, we conducted separate multivariable ordinal regression analyses of perceived difficulty of isolation for each of the retrospectively and prospectively recruited populations. “

“We found similar results when conducting the analysis separately for each of the retro- and prospectively recruited population groups (Supporting information Tables S5a and S5b).”

In ‘abstract- Methods and Findings’ section, lines 33-35 only prospective cohort study [“This population-based prospective cohort study enrolled 1547 adults from the general population with SARS-CoV-2 infection] is mentioned. This mixing confusion is seen at other places also {example: Section on ‘Study design and Population’ (line 108 onwards) “This analysis is based on the Zurich SARS-CoV-2 Cohort study, an ongoing observational, population-based cohort study of individuals in the Canton of Zurich with polymerase chain reaction (PCR) -confirmed SARS-CoV-2 infection”. Please address this issue to clarify more. [No problem if it is so, but should be clarified, remember that this is a scientific/academic document and so all details should be clearly/correctly communicated]

Thank you for pointing this out. We now use “ongoing observational, population-based cohort study” throughout the manuscript (lines 34, 63-64, 110-111, 465, 575)

In the context of information given in lines 144-46 [“To evaluate worries, positive experiences, and difficulties during isolation, we used five-point Likert scale questions with scales ranging from “extremely worried” to “not worried at all”, “strongly agree” to “strongly disagree” and “very difficult” to “very easy”, respectively], please note the following { pasted from one standard textbook on ‘Research Methodology’}:

Whenever response options ranged from 1=strongly disagree to 4=strongly agree (or ranging from 1 (strongly disagree) to 6 (strongly agree) or from 1=very bad to 3=neither good nor bad to 5=very good), while using a ‘Likert’ scale responses, recoding [like strongly disagree=-2, disagree=-1, neutral=0, agree=1, strongly agree=2] may yield correct and meaningful ‘arithmetic mean’ which is useful not only for comparison but has absolute meaning. Application of any statistical test(s) assume that meaning of entity used (mean, SD, etc.) has a particular meaning. Though ‘α’ [alpha] or most other measures of reliability/correlation will remain same, however, use of non-parametric methods should/may be preferred while dealing with data yielded by any questionnaire/score.

Also consider that reported (observed) value of Nagelkerke’s R2 (=0.127) in table-2 is small and table-3 ( R2=0.068 ) is very very small. That the amount of variation explained is all most negligible. From (lines 157-8: Calculation and imputation of DASS-21 scores is described in previously published research from our study and followed official guidance (21).) it seems that the present sample is a sub-sample of some other study. If so, why this fact is not made clear in the beginning? Again, remember that this is a scientific/academic document and so all details should be clearly/correctly communicated.

Thank you for these valuable comments and suggestions. Regarding the re-scaling of the Likert scale to calculate a meaningful arithmetic mean, our main aim was to describe the variation in the responses and to explore associations with baseline characteristics. We agree that the arithmetic mean (with the score centered around zero) would be a useful scale to rank the different statements. However, this was not our main goal. The ordinal logistic regression model (used for Table 2 and Table 3) is parameterized and under the parallel lines and proportional odds assumption.

We agree that the Nagelkerkes R2 is indeed small indicating substantial unexplained variation between individuals which we could be due to factors that were not included in our model. However, it is also important to note that often in analyses relating to behavioral outcomes (for example compliance with isolation recommendation in our case), there may be some inherent variability that cannot be explained, thereby leading to low R2. We have added a statement regarding the low R2 to the limitations section (lines 567-573).

“Last, the methodological approach of analyzing Likert scale items through ordinal regression has its limits and the factors included in our two models explained only a small proportion of the variance in compliance with isolation and perceived difficulty as seen by Nagelkerkes R2. While we found consistent results such as increased difficulty for participants with children and differences between the age groups throughout all analyses and in all the visual presentations, further research is needed to investigate which other factors influence these outcomes.“

Regarding your last comment on the DASS-21 scores and the present study being a sub-sample of the other paper, there has been a misunderstanding which we apologize for. The current study presents the whole cohort. By referencing our other paper, we were referring to the method of calculation of DASS-21 scores as was previously done by our group. We have now made this clear in the methods section (lines 158-159).

“Calculation of the DASS-21 scores followed official guidance (19) and is described in detail in previously published research from our group (21).”

Most of the statistical values/results are not interpreted adequately [example: Odds Ratios in table-3, male Sex (=0.90); Living Alone (=0.60)]. Study has potential, however, I suggest to consider above points.

Thank you for pointing this out. We have added additional interpretation statements in the manuscript (lines 215-219, 296-298, 312-314). 

" … we found evidence that participants with children felt less well informed (OR 1.42, 95% CI 1.12 – 1.79) (Supporting information Table S2). Only 18 participants described themselves as family managers (meaning primary care givers of children and household), and we found weak evidence that family managers also were less well informed (OR 2·11, 95% CI 0·89 – 4·98). “

“ Participants who felt that they were poorly informed about the measures were also more likely to experience difficulties with compliance (OR 3·05, 95% CI 1·91 – 4·87).”

“Furthermore, there was weak evidence for those living alone to experience less difficulties to comply (OR 0·60, 95% CI 0·28 – 1·17) and for those who were self-employed to experience more difficulties to comply (Table 3). “

Moreover, we have changed the formulation of results for more distinct interpretation (lines 270, 292-293, 345-346) and added the individual numbers of the proportions for clarification (lines 322-329).

“Participants who live with children were more worried about stress (173/369, 46·9% vs. 432/1178, 36·6%)…”

“Participants who live with children had twice the odds of experiencing difficulties with adherence to isolation recommendations when compared to those who do not live with children (OR 2·10, 95% CI 1·43 – 3·08).”

“Fig. 3 visualizes the percentage of the population affected by 5 difficult aspects of isolation prespecified in the questionnaire, on Likert scale from very difficult to very easy”

“We observed higher proportions of non-compliance before the positive test result, among those aged 18 to 39 years (70/344, 20·4% compared to 49/448, 10·9% in those aged 40–64 years and 37/313, 11·2% in those over 65 years), living with children (16.0% compared to 13·6% in those who do not live with children), and females (86/565, 15·2% compared to 70/540, 13.0% in males). A higher proportion of participants went to mandatory school or had no degree in the group that were “non-compliant” after the positive test result when compared to the total population (10·5% vs 4·1%) and more were living with children (14/251, 5·6%) when compared to those living without children (24/854, 2.8%).”

Reviewer #2: The authors investigated impacts on the mental health due to the isolation experience in the general population with SARS-CoV-2 infection in Zurich, Switzerland. They found that the proportion of participants affected by depression or anxiety increased during isolation from 10·0% to 17·1% and 9·1% to 17·6%, respectively. They also found that taking care of children increased the difficulty of isolation (OR 2·10, CI 1·43 – 3·08) and risk of non-compliance (OR 1·63, CI 1·05 – 2·53), especially in younger participants. The topic seems to be interesting for researchers and practitioners in public health. In general, this is a well-designed, -analyzed and -written paper. I have some comments to improve this paper.

We thank Reviewer 2 for their insightful and detailed feedback. Line numbers refer to the manuscript without track changes.

1. The description of the results about free text comments, i.e., Results - Free text comments - separate Box – (Line 344 to 435), is redundant for the readers. Results of free text comments should be summarized in one paragraph in the main manuscript. The original description of the results about free text comments should be presented as a supplementary material, if needed.

Thank you for highlighting this point. We have now added a summary paragraph below the comments as you suggested (lines 440 - 461). However, we consider the free text comments as one of the very interesting aspects of this analysis, giving the reader an opportunity to get an impression of the conflicts and problems participants encountered. We would very much like to keep the box but of course leave this decision up to the editor. We present both possibilities in the manuscript for the editor o choose.

“Throughout the free text comments, participants expressed feelings of loneliness and depression as a consequence of their isolation. Isolation was described as “prison”, “solitary confinement” and even “torture” or “punishment” in over 23 comments. Participants described somatic pain due to the lack of physical activity or absence of social contact and described loss of appetite. More than ten participants stated in some form that “isolation was the worst part of their illness”. Participants were most worried about individuals they might have infected, especially loved ones. They also worried about facing stigma and consequences at the workplace. While worries about financial troubles and job security were not frequent, they were highly stressful for those affected by them. Positive comments mainly mentioned spending time with family and relaxing more often. The word “deceleration” appeared frequently. 

The challenge of trying to adhere to isolation recommendations while simultaneously caring for a family with little children was one of the most striking impressions from statements. Participants were especially conflicted when children had to undergo isolation or quarantine themselves: ”completely unsuitable was the isolation for my two children (three and five years old), to lock them in one room for ten days is absurd and not appropriate for little kids”. Families demanded more adjusted measures, help and alternatives for their individual needs. Overall, the call for more and better information, for example about when it was safe to leave isolation and meet people afterwards, was high. There was also a great need for more follow-up and assistance through the government in general, the diagnosing physician or any medical personnel during isolation: “During the whole ten days not one medical professional asked about my health”, “the available assistance did not help us find realizable solutions, we felt unescorted”. These feelings of loneliness and perception of lack of support during this time which made them feel frustrated and helpless. “

2. Clarification of the Figure 1 are needed. The lines of the all graphs are indistinguishable for the readers, e.g., the lines for “Any” and “Severe” in overall with severity, and the lines for “Very severe”, “Severe”, “Moderate” and “Mild” in COVID-19 Symptoms.

Thank you for this comment, we have adapted Figure 1 for better discrimination.

3. There are some typos in the manuscript:

Line 208, 55 participants …

Line 214, that that participants …

Line 252, when isolation has ended Results are …

Line 254, COVID-19 symptoms at infection Out of 1105 …

The errors have all been corrected.

---

## [Decision Letter · Decision Letter 1]

15 Feb 2022

Mental Health of Individuals Infected with SARS-CoV-2 during Mandated Isolation and Compliance with Recommendations - a Population-Based Cohort Study

PONE-D-21-39431R1

Dear Dr. Milo A Puhan,

We’re pleased to inform you that your manuscript has been judged scientifically suitable for publication and will be formally accepted for publication once it meets all outstanding technical requirements.

Kind regards,

Wen-Wei Sung, M.D., Ph.D.

Academic Editor

PLOS ONE

Reviewers' comments:

Reviewer's Responses to Questions

**Comments to the Author**

1. If the authors have adequately addressed your comments raised in a previous round of review and you feel that this manuscript is now acceptable for publication, you may indicate that here to bypass the “Comments to the Author” section, enter your conflict of interest statement in the “Confidential to Editor” section, and submit your "Accept" recommendation.

Reviewer #1: All comments have been addressed

Reviewer #2: All comments have been addressed

2. Is the manuscript technically sound, and do the data support the conclusions?

Reviewer #1: (No Response)

Reviewer #2: (No Response)

3. Has the statistical analysis been performed appropriately and rigorously? 

Reviewer #1: (No Response)

Reviewer #2: (No Response)

4. Have the authors made all data underlying the findings in their manuscript fully available?

Reviewer #1: (No Response)

Reviewer #2: (No Response)

5. Is the manuscript presented in an intelligible fashion and written in standard English?

Reviewer #1: (No Response)

Reviewer #2: (No Response)

6. Review Comments to the Author

Reviewer #1: COMMENTS: All of the comments made on earlier draft by me (and hopefully by other respected reviewers also) were/are attended [though I had suggested minor points and mentioned that the article is excellent by all the means]. I recommend the acceptance as the manuscript now (even earlier I accepted/appreciated the potential of this article) has achieved acceptable level, in my opinion.

Reviewer #2: (No Response)

7. PLOS authors have the option to publish the peer review history of their article (what does this mean?). If published, this will include your full peer review and any attached files.

Reviewer #1: **Yes: **Dr. Sanjeev Sarmukaddam

Reviewer #2: No

---

## [Editor Report · Acceptance letter]

21 Feb 2022

PONE-D-21-39431R1 

Mental Health of Individuals Infected with SARS-CoV-2 during Mandated Isolation and Compliance with Recommendations – a Population-Based Cohort Study 

Dear Dr. Puhan:

I'm pleased to inform you that your manuscript has been deemed suitable for publication in PLOS ONE. Congratulations! Your manuscript is now with our production department. 

Kind regards, 

on behalf of

Dr. Wen-Wei Sung 

Academic Editor

PLOS ONE